

# Application of flux footprint equations from Kljun et al. (2015) to field eddy-covariance systems for footprint characteristics into flux network datasets

Xinhua Zhou[1,3], Zhi Chen[2,*], Ryan Campbell[3], Atefeh Hosseini[3], Tian Gao[1,4], Xiufen Li[5], Jianmin Chu[6], Sen Wu[7], Ning Zheng[7], Jiaojun Zhu[1,4]

[1] Ker Laboratory, Qingyuan Forest CERN, National Observation and Research Station, Liaoning Province, Shenyang 110016, China

[2] Key Laboratory of Ecosystem Network Observation and Modeling, Institute of Geographic Sciences and Natural Resources Research, Chinese Academy of Sciences, Beijing 100101, China

[3] Global Science Program, Campbell Scientific Inc., UT 84321, USA

[4] Qingyuan Forest CERN, National Observation and Research Station, Liaoning Province, Shenyang 110016, China

[5] Department of Agricultural Meteorology, Shenyang Agricultural University, Shenyang 110866, China

[6] Experimental Center of Desert Forestry, Chinese Academy of Forestry, Dengkou 015200, China

[7] Shenzhen Zray-Co Technology Co. Ltd., Shenzhen 518133, China

**Correspondence:** Zhi Chen (chenz@igsnrr.ac.cn)

**Abstract.** Gas fluxes passing through an eddy-covariance (EC) system's measurement volume reflect the outgassing rate of these molecules from an upwind area known as the "flux footprint". While sources/sinks of these molecules may be uniform over a flat field, their spatial contribution to the measured fluxes is not. Thus, understanding the contribution to measured fluxes and the spatial quantification of sources/sinks from the measured fluxes requires footprint analysis. Such analysis

yields flux footprint characteristics, which commonly include upwind maximum footprint location, upwind fetch containing certain percentages of measured flux (70%, 80%, 90%), and the percent of flux from a user-defined upwind fetch of interest. These characteristics are included in the datasets of flux networks such as ChinaFlux, AmeriFlux, and FluxNet. Ideally, the characteristics are calculated in real-time and on-site by EC systems in the field, but this has often not been the case due to the calculations being computationally challenging. For field applications, this study develops the equations and algorithms

for these characteristics from analytical crosswind-integrated flux footprint equations. The development shows that in-field computation is made feasible by the following means: using time-efficient algorithms, taking advantage of the nondimensional nature of the footprint equations of Kljun et al. (2015), implementing practical limits on numerical integration, and developing a differential-based estimation of boundary layer height for each EC interval. Accuracy of in-field calculations is maintained by the selection of footprint equations based on boundary-layer conditions and considerations of integration methods and computation techniques. This computational approach may also be applied to

footprint analyses over complex terrain, nonuniform sources/sinks, or in cases where other footprint equations are used. The most popular application of footprint analysis is to optimize the EC sensor height for maximization of measured fluxes from an area of interest. This optimization using the nondimensional footprint equations is discussed, which leads to a practical





methodology. This work serves as a technical reference for users or developers of EasyFlux programs, widely used in
Campbell Scientific EC systems globally.

## 1 Introduction

An eddy-covariance system for flux measurements, including a gas analyzer (e.g., an infrared $CO_2$–$H_2O$ analyzer) and three-dimensional (3D) sonic anemometer, is mounted at its measurement height $z_m$ on a field tower (Munger et al., 2012). The gas analyzer and sonic anemometer are configured for their sensing surfaces to enclose the outmost boundary of the
"measurement volume" (see IRGASON in Fig. 1a), through which passive gas, sensible heat, and momentum fluxes are measured. These measured passive gas fluxes (e.g., $CO_2$) through the measurement volume are stochastically transferred by boundary-layer turbulent flows (Horst and Weil, 1992) from their sources or to their sinks over an area called the flux footprint. As such, atmospheric conditions and the spatial relation of the measurement volume to the sources/sinks determine the molecular number of a measured passive gas flux from or to a particular unit area over the flux footprint field. In other
words, the flux contribution varies spatially (Fig. 1a). This is the case even when the rate of source emission or sink absorption is spatially uniform (Hsieh et al., 2000). However, given that in common instances this rate may be spatially nonuniform, for practical cases over heterogeneous or sporadic sources/sinks, flux footprint equations are needed for evaluation of sources/sinks (Leclerc and Foken, 2014).

In a boundary-layer turbulent flow field, a flux footprint equation $f(x, y)$ is spatially defined in a wind coordinate system,
with $x$ in a direction against streamwise wind, $y$ horizontally across streamwise wind, and $z$ orthogonal to $x$ and $y$ (Fig. 1 in Schmid, 1994). To easily relate a wind coordinate $(x, y)$ to its ground location, the horizontal coordinate of a flux tower base is assigned as the origin $(0, 0)$ in both the wind and ground coordinate systems. In this way, given a wind direction in reference to the ground location of a flux tower, any location at $(x, y)$ can be trigonometrically related to its ground location. In the wind coordinate system, $f(x, y)$ is understood to be a probability distribution of contributions (Kormann and Meixner,
2001) from the spatially uniform sources/sinks of a passive gas over a topographically flat field to its turbulent flux passing through the "measurement volume" of an eddy-covariance flux system. Thus, for uniform sources/sinks of a passive gas over a flat fetch, a footprint value at a particular ground location indicates a relative contribution of passive gas from this location to the measurement volume centered at $(0, 0, z)$, where $z$ is the aerodynamic height equal to $z_m$ minus $d$ (zero-plane displacement height). The greater the footprint value, the more contribution from that location.
The flux footprint $f(x, y)$ can be integrated across wind (i.e., along $y$) to yield a crosswind-integrated flux footprint $f_y(x)$, resulting in a skewed bell-shape curve along the $x$-axis only (Fig. 1a). The curve represents a probability distribution. A common convention in literature is to present flux footprints for positive fluxes, where a gas is emitted from its upwind sources. However, footprints apply to negative fluxes with sinks as well. Regardless of the flux direction, all gas molecules in boundary-layer turbulent flows are transported through a stochastic process (Lumley and Panofsky, 1964; Horst, 1979),
and accordingly, the flux footprint curve for measured flux from upwind sources along the positive $x$ (Fig. 1) should be





**Figure 1:** A flux footprint equation is displayed as a crosswind-integrated flux footprint, where the curve along the $x$-axis shows the contribution of molecules originating at a particular value of $x$ and for all values of $y$.  a. Crosswind-integrated flux footprint in a case where $CO_2$ flux sources are known to be uniform and emitted at a rate of 10 $CO_2$ molecules m$^{-2}$ s$^{-1}$ under convective conditions. The shape of the curve is affected by changes in sensor aerodynamic height $z$ (*i.e.*, height of center of measurement volume minus the zero-plane displacement height (panel b) and by changes in atmospheric boundary-layer stability, as determined from the Monin-Obukhov length $L$ and friction velocity $u_*$ (panel c). All curves in this figure are computed using Eq. (22) in Kormann and Meixner (2001). For these curves, unless noted inside panels, $z$ is 6 m, $L$ is −650 m, wind speed is 5 m s$^{-1}$, $u_*$ is 0.3 m s$^{-1}$, and the van Karman constant is 0.41.




symmetric with its counterpart for measured fluxes to downwind sinks along the negative $x$. This symmetry should be true

around the $z$-axis in the $x$ and $y$ domain, too (Schmid, 1997). Because of this symmetry and for simplification, only the cases of upwind flux footprint for a positive flux from its upwind sources are conventionally presented in literature (e.g., Schmid, 2002). Such a conventional presentation is followed by this study for figures, equations, and algorithms.

As shown in Fig. 1 for upwind flux footprint curves, even in cases where gas flux over a vast flat field is uniform, the flux footprint varies with upwind distance away from the measurement volume. It is also shaped by the aerodynamic height

of the measurement volume (Fig. 1b, Hsieh et al., 2000; Raupach et al., 1991) and by boundary-layer conditions related to thermodynamic stratifications in air flows (Fig. 1c, Kormann and Meixner, 2001).

As a probability distribution, $f(x, y)$ can be used to derive a mean of passive gas sources/sinks $Q(x, y)$ over a 2-dimensional (2D) field (Snedecor and Cochran, 1989) because both are related to the flux through the measurement volume $F(0, 0, z)$ (Kormann and Meixner, 2001):

$$F(0,0,z) = \int_{\Re} Q(x,y) f(x,y) dx dy \qquad\qquad (1, \text{model})$$

where $\Re$ denotes an integration domain. Indeed, $f(x, y)$ may be thought of as a transfer function of the gas flux of $Q(x, y)$ over an extended 2D field to the flux at the measurement volume $F(0, 0, z)$ (Kljun et al., 2015). Accordingly, although developed based on horizontally uniform sources/sinks of a passive gas, $f(x, y)$ is also applicable to the description of the transfer process of passive gas flux signals from nonuniform sources/sinks, represented by $Q(x, y)$ (see Chapter 8 in Leclerc

and Foken, 2014; Göckede et al., 2004).

The ultimate objective from a measured flux $F(0, 0, z)$ is to evaluate $Q(x, y)$ over the ecosystems targeted for measurement. For horizontally nonuniform sources/sinks over flat terrain, $Q(x, y)$ varies with $x$ and $y$. In this case, $f(x, y)$ is imperative for $Q(x, y)$ to be evaluated from $F(0, 0, z)$, which is an advanced application of $f(x, y)$ still under development (Leclerc and Foken, 2014). In cases where $Q(x, y)$ is constant for horizontally uniform sources/sinks of measured gas over

flat terrain, $F(0, 0, z)$ must be representative to this constant due to the right side of model (1) to be this constant because the integration of $f(x, y)$ alone over its full domain is equal to a unit (Snedecor and Cochran, 1989). For most flux measurements, this scenario is assumed, thus, for scenarios where $Q(x, y)$ is constant, $f(x, y)$ is less significant.

However, modern flux network datasets, most of which are from sites of assumed horizontally uniform sources/sinks over flat terrain, report footprint characteristics including the upwind maximum footprint location

(*FETCH_MAX, i.e.,* distance at which the sources/sinks contribute most to the measured flux) and the upwind fetch within which the sources/sinks contribute a given percentage to the measured flux (e.g., *FETCH_*70 for 70%, *FETCH_*80 for 80%, and *FETCH_*90 for 90%). Additionally, EasyFlux outputs the interest fetch (*FP_FETCH_INTRST,* i.e., the integrated flux contribution from a defined fetch of interest). These footprint characteristics are increasingly becoming essential variables in many datasets from international networks (e.g., AsiaFlux, https://www.asiaflux.net; FLUXNET, http://fluxnet.org; and



ICOS, www.icos-infrastructure.eu), national networks (e.g., AmeriFlux, http://ameriflux.lbl.gov and ChinaFlux, http://www.chinaflux.org), regional networks (e.g., NYS Mesonet, https://www.nysmesonet.org/), and individual eddy-covariance flux stations. In these networks and stations, thousands of Campbell Scientific eddy-covariance flux systems have been deployed based on instruments such as the IRGASON (integrated open-path infrared $CO_2$–$H_2O$ analyzer and 3D sonic anemometer), CPEC300 series (EC155 closed-path infrared $CO_2$–$H_2O$ analyzer with CSAT3A), and TGA (Trace Gas

Analyzer) with CSAT3B (Campbell Scientific Inc., UT, USA). Each of these systems is controlled and measured by a datalogger (e.g., CR6, CR1000X, or Granite9, Campbell Scientific Inc., UT. USA), which also processes, transfers, and stores data.

Each datalogger operates a program from the EasyFlux series, which handles instructions for system control, field measurements, and data transfers (e.g., to FTP site or Campbell Cloud). And most importantly, the EasyFlux program

processes raw high-frequency (e.g., up to 20 Hz) measurements into fully corrected fluxes every user-specified output interval (e.g., 30 min). Other required variables, including footprint characteristics from the analytical crosswind-integrated flux footprint equations of Kormann and Meixner (2001) or Kljun et al. (2004; 2015), are also output each interval. The recent implementation of the equations from Kljun et al. (2015) is a new update, as previously the equations from Kljun et al. (2004) were used. The applicability of this update is important because of its consideration of various boundary-layer

stabilities. Due to this advancement, EasyFlux series programs released hereafter are programmed Kljun et al. (2015) as its default option for flux footprint characteristics, although Kormann and Meixner (2001) for these is still available as an alternative.

The primary goal of this study is to develop efficient algorithms for applying Kljun et al. (2015) in a datalogger, thus allowing for in-field computations of footprint characteristics every output interval. And since the resulting algorithms

have been implemented in recent versions of EasyFlux datalogger programs, this paper also serves as a reference for the users and developers of Campbell Scientific eddy-covariance flux stations who wish to know technical details about the flux footprint characteristic outputs. But first, to comprehend the algorithms related to Kljun et al. (2015), we briefly summarize the development of their flux footprint equations.

## 2 Brief the development of flux footprint equations by Kljun et al. (2015)

Using the backward Lagrangian stochastic particle dispersion model (LPDM-B), Kljun et al. (2015) simulated the flux footprint for a vast range of values for $z$, going between 1 and 1,000 m in boundary-layer conditions ranging from strongly convective through neutral to strongly stable, and a large range of values for roughness length $z_0$, including values for sparse forest canopies (Fig. 1 in Kljun et al., 2015). The vast range in flux footprint sizes (e.g., up to 270 km for only 80% footprint) manifests that it is not practical for a limited number of analytical $f(x, y)$ equations to meet the needs for all boundary-layer

flow fields at field scales. However, if the variables in $f(x, y)$ are made dimensionless, $f(x, y)$ could be independent of the dimensions of boundary-layer flow fields.





Ideally, $f(x, y)$ contours for all flow fields converge to a single shape or narrow ensemble, regardless of the magnitude of the field dimensions or the boundary-layer conditions. Thus, the single shape may be regarded as dimensionless and applicable to any field size and in any condition of atmospheric stability. With this aim, Buckingham $\Pi$ dimensional analysis (Stull, 1988) is an approach of Kljun et al. (2015) to formulate the universal model for this contour. The data from the LPDM-B simulations include a vast range of boundary-layer flows, as characterized by the combinations of $z$, $z_0$, and boundary-layer stabilities, and thus are a good source of statistical samples for determination of the model parameters.

## 2.1 Buckingham $\Pi$ dimensional analysis of flux footprint

In a case where $\mathfrak{R}$ is infinitely small, model (1) can be written as

$$F(0, 0, z) = f(x, y)Q(x, y)\Delta x \Delta y , \tag{2}$$

which is equivalent to

$$f(x, y)\frac{F(0, 0, z)}{Q(x, y)\Delta x \Delta y} , \tag{3}$$

where $F$ and $Q$ have the same units given in mass/molecules m$^{-2}$ s$^{-1}$. Accordingly, $f(x, y)$ has units of m$^{-2}$ since that would be the units of $1/\Delta x \Delta y$ if $x$ and $y$ are in m

The flux footprint characteristics in AmeriFlux (2018) datasets include *FETCH_MAX*, *FETCH_70*, *FETCH_80*, and *FETCH_90*, which are all measured in terms of an upwind fetch in m. Within a fetch, the relative contribution to the measured gas flux from horizontally uniform sources/sinks of a passive gas over a flat field is an accumulation of $f(x, y)$ after integration across wind, defined as $f_y(x)$. For the computations of flux footprint characteristics, as addressed in this study, only $f_y(x)$ is needed, although $f(x, y)$ may still be desired for flux footprint maps in two dimensions (Kormann and Meixner, 2001; Kljun et al., 2004; 2015). If the independent dispersion of a passive gas across wind is described by $D(y)$, $f_y(x)$ forms a 2D flux footprint $f(x, y)$ given as (Horst and Weil, 1992):

$$D(y)f_y(x) = f(x, y) . \tag{4}$$

Although the explicit equation of $D(y)$ is omitted here, it is a probability distribution (Pasquill and Smith, 1983) whose integration over $y$ is equal to a unit. Because $f_y(x)$ is not dependent on $y$, the integration of Eq. (4) with respect to $y$ yields

$$f_y(x) = \int_{-\infty}^{\infty} f(x, y)dy \tag{5}$$

Thus $f_y(x)$ has the same dimension as $f(x, y)dy$, which is m$^{-1}$. Its dimension is fundamental to nondimensionalization of $f_y(x)$ using Buckingham $\Pi$ dimensional analysis (Stull, 1988).

## 2.2 Buckingham $\Pi$ dimensionless combinations

$f_y(x)$ is a function of upwind fetch ($x$ in m), varying with $z$ in m, mean wind speed $\overline{u}(z)$ in m s$^{-1}$, friction velocity $u_*$ in





m s$^{-1}$, $z_0$ in m, and the planetary boundary layer height $h$ in m. In sections 3 and 4 of Kljun et al. (2015), these dimensional variables are used for their Eq. (4) to (14) to formulate each dimensionless combination $\Pi$ that will be used to nondimensionalize $f_y(x)$, as briefed below.

The first choice is $z$ because both the extent and magnitude of the footprint are most strongly dependent on it (Hsieh et al., 2000). The higher the measurement volume at $z$, the farther the footprint stretches along the upwind fetch (Fig 1b).

Accordingly, the independent fetch variable $x$ of $f_y(x)$ should be inversely nondimensionalized by $z$ as combination $\Pi_1$:

$$\Pi_1 = \frac{x}{z} \ . \tag{6}$$

Another effect is that the $f_y(x)$ curve on average has a lower value when $z$ is higher and the footprint is stretched along the upwind fetch, (Schmid, 1997). Therefore, $f_y(x)$ should be positively nondimensionalized by $z$ as combination $\Pi_2$:

$$\Pi_2 = z f_y(x) \ . \tag{7}$$

According to a common finding that turbulent fluxes decline approximately linearly through the planetary boundary layer from surface value to zero at $h$ (e.g., Stull 1988), $z$ and $h$ can be nondimensionalized as combination $\Pi_3$:

$$\Pi_3 = 1 - \frac{z}{h} \ . \tag{8}$$

As a transfer function in turbulent boundary layer flows, the flux footprint is directly affected by $\overline{u}(z)$, $u_*$, and $z_0$. Well-known nondimensional wind shear $\phi_m$ explicitly and implicitly includes these three variables (Kaimal and Finnigan, 1994),

given by:

$$\phi_m = \frac{kz}{u_*} \frac{\partial \overline{u}(z)}{\partial z}, \tag{9}$$

where $k$ is the von Karman constant (0.41). If the derivative is replaced by its approximation at $z$, $\phi_m$ becomes

$$\phi_m \approx \frac{kz}{u_*} \frac{\overline{u}(z) - \overline{u}(z_0 + d)}{z - (z_0 + d)} \approx k \frac{\overline{u}(z)}{u_*} \ . \tag{10}$$

From Kaimal and Finnigan (1994) and Högström (1996), $\phi_m$ is influenced by $z_0$ because:

$$k \frac{\overline{u}(z)}{u_*} = \ln\left(\frac{z - d}{z_0}\right) - \psi_m \tag{11}$$

where $\psi_m$ is the integrated form of nondimensional wind shear (Kaimal and Finnigan, 1994), which accounts for the effects of stability ($z/L$, where $L$ is Monin-Obukhov length). If $\phi_m$ is thought of as nondimensional wind speed at $z$, reflecting a combined effect of $u$, $u_*$, and $z_0$, it follows to use it as combination $\Pi_4$:

$$\Pi_4 = k \frac{\overline{u}(z)}{u_*} \tag{12}$$

Unlike Kljun et al. (2004) which uses $z_0$ explicitly, combination $\Pi_4$ here includes $z_0$ implicitly.



### 2.3 Nondimensional upwind fetch ($X^*$)

The footprint of the measurement volume of an eddy0covariance flux systems at a given $z$ extends farther when $h$ is higher (i.e., positively proportional to $\Pi_3$) and shrinks when wind is stronger (i.e., inversely proportional to $\Pi_4$). Accordingly, Kljun et al. (2015) formed nondimensional upwind fetch as:

$$X^* = \Pi_1 \Pi_3 \Pi_4^{-1} = \frac{x}{z}\left(1 - \frac{z}{h}\right)\left(k\frac{\overline{u}(z)}{u_*}\right)^{-1} \tag{13}$$

### 2.4 Nondimensional crosswind-integrated flux footprint ($F_y^*$)

Because the integration of the flux footprint over its full range is always equal to 1, individual footprint values are on average lower when the footprint has a longer range, and higher when the footprint has a shorter one. Therefore, $\Pi_3$ and $\Pi_4$ interact inversely. Accordingly, the nondimensional crosswind-integrated flux footprint can be formulated as:

$$F_y^* = \Pi_2 \Pi_3^{-1} \Pi_4 = f_y(x)z\left(1 - \frac{z}{h}\right)^{-1}\left(k\frac{\overline{u}(z)}{u_*}\right). \tag{14}$$

Even when $f_y(x)$ extends to very long ranges as shown in Fig. 1 of Kljun et al. (2015), $F_y^*$ versus $X^*$ converges to an ensemble of nondimensionalized crosswind-integrated flux footprints very similar in curve shape, peak location, and fetch extent (see Fig. 2 of Kljun et al. 2015).

### 2.5 Formulation and parameterization for $F_y^*$

For a given range of boundary-layer stabilities, the convergence of $F_y^*$ versus $X^*$ to a narrow ensemble provides the basis to formulate a universal model fitted to the ensemble of LPDM-B results. Additionally, Kljun et al. (2015) chose to describe the relationship of $F_y^*$ to $X^*$ using the product of a power function of $X^*$ and an exponential function of $X^*$ (see their Fig. 2). The product formulates a universal model for the non-dimensional crosswind-integrated flux footprint:

$$F_y^*(X^*) = a\left(X^* - d_0\right)^b \exp\left(-\frac{c}{X^* - d_0}\right), \tag{15, model}$$

where $a$, $b$, $c$, and $d_0$ are parameters, and the subscript 0 is used to avoid confusion between the fourth parameter and the zero-plane displacement height, conventionally denoted by a $d$ in boundary-layer meteorology. Because model (15) is a probability distribution, its four parameters satisfy a constraint where the integral of $F_y^*(X^*)$ over the $X^*$ domain must be unity:

$$\lim_{\delta_0 \to d_0} \int_{\delta_0}^{+\infty} F_y^*(X^*)dX^* = 1, \tag{16}$$

where $\delta_0$ is the lower limit of integration. Using an alternative variable,



$$t = \frac{c}{X^* - d_0} \, ,$$ (17)

the integral of the right side of model (15) can be related to the Gamma function (Nemes, 2010) as

$$ac^{b+1} \int_0^{+\infty} t^{-b-2} \exp(-t) dt = ac^{b+1} \Gamma(-b-1) = 1$$ (18)

With this constraint, the parameters in model (15) were statistically estimated using the data from LPDM-B simulations after
nondimensionalization. With a set of estimated parameters, model (15) was developed into a non-dimensional crosswind-integrated flux footprint equation.

As shown in Fig. 2 of Kljun et al. (2015), this equation represents the flux footprint across all field scales, with model (15) shown as the universal framework. The goodness-of-fit of this single $F_y^*$ equation for the ensemble of nondimensionalized flux footprints for all simulated measurement heights, stability conditions, and roughness lengths is
evidenced from the model performance metrics in Table 3 of Kljun et al. (2015). The fit can be improved even more if model parameters are optimized as two sets as shown in Table A1 of Kljun et al. (2015), each of which represent $F_y^*$ under convective ($z/L < 0$) or neutral/stable ($z/L \geq 0$) boundary-layer conditions. Thus, a pair of equations are formulated as a set for $F_y^*$:

$$F_y^*(X^*) = \begin{cases} \dfrac{2.930}{\left(X^* + 0.107\right)^{2.285}} \exp\left(-\dfrac{2.127}{X^* + 0.107}\right) & \dfrac{z}{L} < 0 \\[2em] \dfrac{1.472}{\left(X^* - 0.169\right)^{1.996}} \exp\left(-\dfrac{1.480}{X^* - 0.169}\right) & \dfrac{z}{L} \geq 0 \end{cases}$$ (19)

This set of analytical crosswind-integrated flux footprint equations are adopted into the EasyFlux series of programs.

**3  Applications of nondimensional crosswind-integrated flux footprint equations**

In the EasyFlux series, the nondimensional crosswind-integrated flux footprint equations for $F_y^*(X^*)$ as shown in Eq. (19) are adopted to estimate the footprint characteristics over a flat field with horizontally uniform sources/sinks of passive gases. For example, $FETCH\_70$ is found by integrating Eq. (19) from a starting limit to $X_{70}^*$, It is the upper integration limit that
results in a cumulative footprint probability of 0.7. $X_{70}^*$ is converted to field scale units (e.g., meters) using Eq. (13). Similarly, $FETCH\_80$ and $FETCH\_90$ may be found. For $FT\_FETCH\_INTRST$, which is the percentage of measured flux attributable to the area within a user-defined fetch distance, $fetch\_intrst$. Also, through Eq. (13), this field distance is converted to $X_{\mathrm{intrst}}^*$, to which Eq. (19) is integrated from its starting limit, yielding $FT\_FETCH\_INTRST$.



Since the integrations described above can be computationally intensive and difficult to do in the field, the following

sections discuss approaches for calculating the footprint characteristics that eliminate or reduce in-field numerical integration.

### 3.1 FETCH_MAX

$F_y^*\left(X^*\right)$ yields skewed bell-shaped curves with respect to $X^*$ (Fig. 2). The location of the maximum in terms of

nondimensional upwind fetch $X_{max}^*$ is given by Eq. (20) of Kljun et al. 2015 (see derivation in Appendix A):

$$X_{max}^* = d_0 - \frac{c}{b} \qquad (20)$$

Its values for two ranges of atmospheric conditions are computed and shown in Table 1.

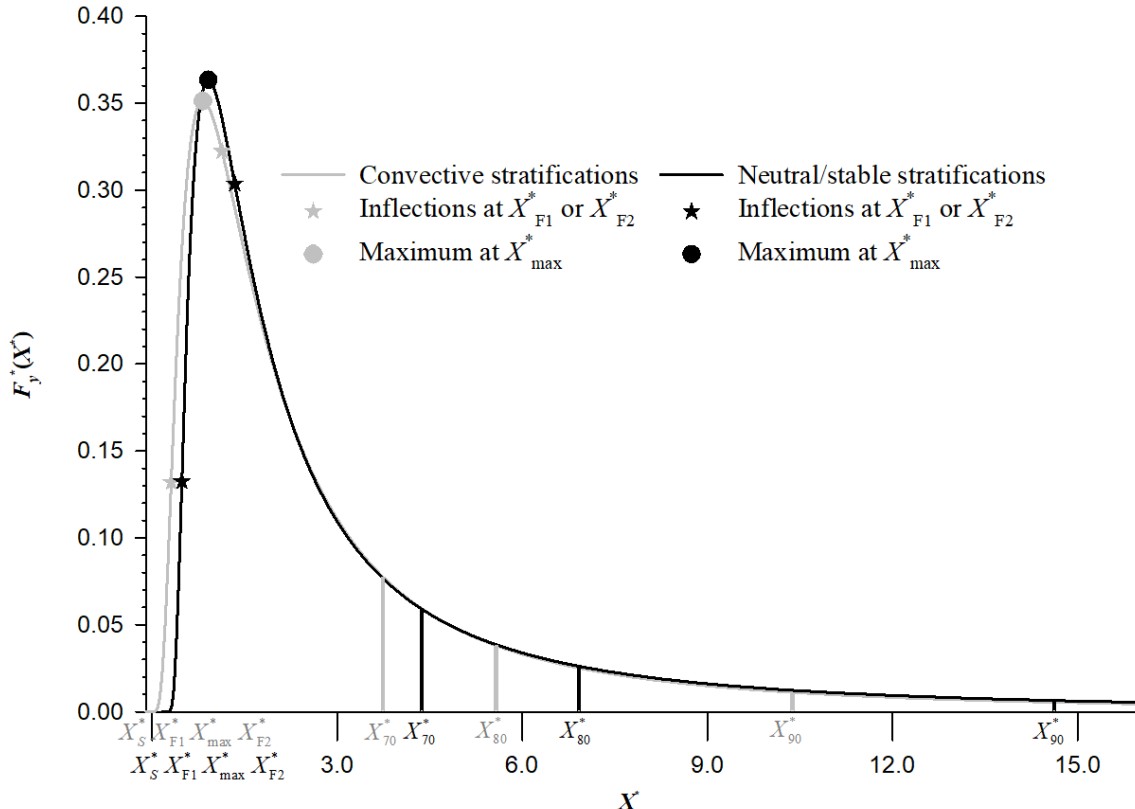

**Figure 2:** Nondimensional crosswind-integrated flux footprint $F_y^*(X^*)$ as a function of nondimensional upwind fetch $X^*$. A vertical bar at $X_p^*$, where subscript $p$ indicates the percent of 70, 80, or 90, is the boundary at which the integration of $F_y^*(X^*)$, as shown in Eq. (19), from its starting limit $X_S^*$ is equal to $p\%$. $X_{max}^*$ is the location of the maximum value of $F_y(X^*)$. $X_{F1}^*$ and $X_{F2}^*$ are the 1st and 2nd inflections

on a $F_y(X^*)$ curve.





**Table** 1. The 1st inflection $X_{F1}^*$, maximum $X_{max}^*$, and 2nd inflection $X_{F2}^*$ on a nondimensional crosswind-integrated flux footprint curve $F_y^*(X)$ along the nondimensional upwind fetch $X^*$ ($z$ is aerodynamic height for flux measurements and $L$ is Monin-Obukhov length.)

| Atmospheric stability | $X_{F1}^*$ | $X_{max}^*$ | $X_{F2}^*$ |
|---|---|---|---|
| Convective $z/L < 0$ | 0.31026689[a] | 0.82385339 | 1.3374399 |
| Neutral/stable $z/L \geq 0$ | 0.48210189 | 0.91048297 | 1.3388640 |

[a] For each number, at least eight digits are kept for computations in single precision.

In Eq. (13), when $X^*$ equals $X_{max}^*$, $x$ becomes *FETCH_MAX* and is given by:


$$FETCH\_MAX = kX_{max}^* z \frac{\overline{u}(z)}{u_*}\left(1-\frac{z}{h}\right)^{-1}. \tag{21}$$

Over an averaging interval (e.g., 30 min) in a field eddy-covariance flux system, $\overline{u}(z)$ and $u_*$ are derived from its wind measurements, $h$ can be either directly measured using a Lidar Ceilometer (e.g., SkyVue Pro, Campbell Scientific Inc., 2025) or alternatively estimated from its measurements using the algorithms in Appendix B, and $z$ is the aerodynamic height which is calculated from $z_m$ minus $d$, where $d$ in a field eddy-covariance flux system can be entered by the system user as the first

choice whereas it is automatically estimated inside the system from the height of canopies around the flux tower (Rosenberg et al., 1983; Oke, 1987; Kaimal and Finnigan, 1994). The sensor measurement height $z_m$ and canopy height are also included among the station variables whose values are set by the system user onto an EasyFlux program before or while an eddy-covariance system is running (Campbell Scientific Inc., 2022).

### 3.2 *FP_FETCH_INTRST*

*FP_FETCH_INTRST* is the cumulative footprint probability within a specified upwind fetch, *fetch_intrst*. In EasyFlux series, *fetch_intrst* is one of the so-called station variables that are entered by a system user into the EasyFlux program before or while the station is running. Using Eq. (13), it can be used to compute its corresponding nondimensional form $X_{intrst}^*$. In this equation, at $x$ equal to *fetch_intrst*, $X^*$ is $X_{intrst}^*$ and given by:

$$X_{intrst}^* = \frac{fetch\_intrst}{z}\left(1-\frac{z}{h}\right)\left(k\frac{\overline{u}(z)}{u_*}\right)^{-1}. \tag{22}$$

Accordingly, the footprint percentage of measured passive gas flux within *fetch_intrst* around a flux tower is an integration of $F_y^*(X^*)$ with respect to $X^*$ from its starting limit $X_S^*$ (near the flux tower) to $X_{intrst}^*$:

$$FP\_FETCH\_INTRST = 100\int_{X_S^*}^{X_{intrst}^*} F_y^*(X^*)dX^*, \tag{23}$$



where $X_S^*$ can be set to $d_0 + 10^{-7}$ because $F_y^*(X^*)$ is valid only when $X^*$ is greater than $d_0$ and, in the 7th significant digit after decimal (i.e., single precision expression), $d_0 + 10^{-7}$ is a number near the smallest precise number greater than $d_0$.

**3.2.1    Computation considerations**

As explicitly expressed in Eqs. (19) and (23), $F_y^*(X^*)$ may be numerically integrated at discrete, incremental values of $X^*$, starting at $X_S^*$ and increasing until $X_{intrst}^*$ is reached. The accuracy of numerical integration depends on the resolution of increments in $X^*$. The smaller the increment, the higher resolution and greater accuracy the result (Burden et. al., 2016). However, for a given range of $X^*$, smaller increments increase the number of iterations for numerical integration, which adds more computational loads to a microprocessor of an in-field computation module, such as a CR6 or CR1000X datalogger (Campbell Scientific Inc. UT, USA), commonly used inside of an eddy-covariance flux system. Thus, the integration for field applications must be optimized to ensure integration accuracy with a minimized computational load.

As shown in Fig. 2 for Eq. (19), $F_y^*(X^*)$ has four identified turning points: the starting limit at $X_S^*$, the maximum at $X_{max}^*$, and the bilateral inflection points at $F_{F1}^*$ and $X_{F2}^*$ as well. Since the flux footprint curve changes more rapidly around these points, the accuracy of numerical integration would include less uncertainties if these points were located at the boundaries for segments or zones of integration (Burden et. al., 2016). Additionally, as compared to the right tail of the flux footprint curve, the curve across the three zones from $X_S^*$ to $F_{F1}^*$, $F_{F1}^*$ to $X_{max}^*$, and $X_{max}^*$ to $X_{F2}^*$ is steeper in slope or changes more dramatically. Since one of the two end points of each zone is an inflection point, these zones will be called inflection zones for the purposes of this study.

Within a zone, an increment for numerical integration should be small for greater accuracy, and $X_S^*$, $F_{F1}^*$, $X_{max}^*$, and $X_{F2}^*$ are used as boundaries. Beyond $X_{F2}^*$, an integration increment may be extended, creating lower resolution but reducing computations. As previously noted, $X_S^*$ is defined based on $d_0$, which is a parameter in model (15) and used as a constant in Eq. (19). $X_{max}^*$ is given by Eq. (20). Derived in Appendix A, the first inflection point is located at

$$X_{F1}^* = \frac{-c\left[1 - (1-b)^{-\frac{1}{2}}\right]}{b} + d_0 \qquad (24)$$

and the second one is located at

$$X_{F2}^* = \frac{-c\left[1 + (1-b)^{-\frac{1}{2}}\right]}{b} + d_0 \qquad (25)$$

For $F_{F1}^*$ and $X_{F2}^*$, their computed values are shown in Table 1, and their locations on the footprint curve are shown in Fig. 2.



### 3.2.2 Algorithm

As discussed previously, Eq. (22) is used to nondimensionalize the upwind fetch of interest to $X^*_{\text{intrst}}$ and the numerical integration of Eq. (23) to $X^*_{\text{intrst}}$ yields the footprint fraction of measured flux sourced from the upwind fetch of interest. For the integration, we choose the Composite Simpson's Rule (Burden et al., 2016). Depending on $X^*_{\text{intrst}}$, the integration can cover, from left to right as shown in Fig. 2, one to three full inflection zones unless $X^*_{\text{intrst}} < X^*_{F1}$. To reduce the uncertainties in the accuracy over the range of integration, $F^*_y(X^*)$ in Eq. (23) should be integrated at higher resolution with smaller increments over these zones, but beyond them (i.e., $X^* > X^*_{F2}$), the integration can be performed at lower resolution with an increased size of increments.

One thousand increments of $X^*$ within an inflection zone are considered adequate, with increments smaller than $5.14 \times 10^{-4}$ (Table 2). For the inflection zone in which $X^*_{\text{intrst}}$ is located, only the portion of the zone up to $X^*_{\text{intrst}}$ is numerically integrated in the field. In this way, the computational load for *FP_FETCH_INTRST* can be controlled to its minimum so that in-situ outputs are possible while the full infection zones within $X^*_{\text{intrst}}$ are numerically integrated in a lab at high resolution as shown in Table 2.

Within an inflection zone that $X^*_{\text{intrst}}$ is located and up to the value of $X^*_{\text{intrst}}$ the resolution in Table 2 is used. For inflection zones lower than the zone in which $X^*_{\text{intrst}}$ is located, no integration is required, as calculated constants for cumulative footprint in each zone may be used (see Table 2). In cases where $X^*_{\text{intrst}}$ is beyond the second inflection point, the integration increment between $X^*_{F2}$ and $X^*_{\text{intrst}}$ is determined by $\left(X^*_{\text{intrst}} - X^*_{F2}\right)/n$ where $n$ is typically 1000 or less in order to limit the time needed for computation. The number of increments $n$ for the lower resolution depends on the computation capacity of the microprocessor in a field eddy-covariance flux system. It should be noted that the numerical integration calculations also rely on inputs from real-time eddy-covariance sensor measurements, because as shown by Eq. (22), the evaluation of $X^*_{\text{intrst}}$ requires $\overline{u}(z)$, $u_*$, and $h$, which are calculated from in-field high-frequency measurements.

### 3.2.3 Example

Given that an upwind fetch of interest is 500 m, $z$ equals 5 m, and the conditions for scenario 3 in Table 1 of Kljun et al (2015) ($L=-650$ m, $u_* = 0.30$ m s$^{-1}$, and $h = 1,200$ m) with $\overline{u}(5)$ equal to 4.00 m s$^{-1}$, $X^*_{\text{intrst}}$ from Eq. (22) is 18.216463. Because $X^*_{\text{intrst}}$ is greater than $X^*_{F2}$ (Table 1), using Eq. (23), the flux footprint percentage within this upwind fetch to the flux tower can be evaluated by:

$$FP\_FETCH\_INTRST = 100\int_{X^*_S}^{X^*_{F2}} F^*_y\left(X^*\right)dX^* + 100\int_{X^*_{F2}}^{X^*_{\text{intrst}}} F^*_y\left(X^*\right)dX^* \qquad (26)$$





**Table** 2. The flux footprint values in inflection zones (Fig. 2) and their cumulative flux footprint values from the starting value of nondimensional upwind fetch $X^*$, denoted by $X_S^* = d_0 + 10^{-7}$ where $d_0$ is a parameter of nondimensional crosswind-integrated flux footprint equation $F_y^*(X^*)$ as shown in Model (15) and Eq. (19). The flux footprint values are numerically integrated for each inflection zone using the Composite Simpson's Rule on a $F_y^*(X^*)$ curve (Fig. 2). $X_{F1}^*$ is the 1st inflection ahead of $X_{max}^*$ (the maximum location) and $X_{F2}^*$ is the 2nd inflection behind $X_{max}^*$. $L$ is Monin-Obukhov length and $z$ is the aerodynamic height for measurements.

| Atmospheric stability | Zone | Ending | $X_{F1}^*$ | $X_{max}^*$ | $X_{F2}^*$ |
|---|---|---|---|---|---|
| | | Range | $X_S^* \sim X_{F1}^*$ | $X_{F1}^* \sim X_{max}^*$ | $X_{max}^* \sim X_{F2}^*$ |
| Convective $z/L < 0$ | | Integration resolution | $4.1726699 \times 10^{-4\,a}$ | $5.1358650 \times 10^{-4}$ | |
| | | Zone footprint % | 1.1321783 | 14.605774 | 16.788529 |
| | | Cumulative footprint % [b] | | 15.737952 | 32.526482 |
| Neutral/stable $z/L \geq 0$ | | Integration resolution | $3.1310199 \times 10^{-4}$ | $4.2838107 \times 10^{-4}$ | |
| | | Zone footprint % | 0.87452260 | 12.597578 | 14.546249 |
| | | Cumulative footprint % | | 13.472100 | 28.018350 |

[a] For each number, eight digits are kept for significance of computations in single precision at least.
[b] Cumulative footprint in each zone column is the integration of $F_y^*(X^*)$ from $d_0 + 10^{-7}$ to the ending boundary of this zone.

The 1st term on the right side of this equation was evaluated in Table 2 as a constant 32.526482 %. For field applications, Eq. (26) for this case can be expressed as:

$$FP\_FETCH\_INTRST = 32.526482\% + 100 \int_{X_{F2}^*}^{18.216463} F_y^*(X^*)\,dX^* \qquad (27)$$

Thus, in the field, numerical integration is required only on the 2nd term on the right side. If $n$ is 1,000, the size of increments in $X^*$ for numerical integration is given by:

$$\Delta X^* = \frac{X_{intrst}^* - X_{F2}^*}{n} = \frac{18.216463 - X_{F2}^*}{1000} = 1.6879023 \times 10^{-2} \qquad (28)$$

Appendix C shows the algorithms used for numerical integrations of Eqs. (27) and (28) using the Composite Simpson's Rule. In this example, after integration $FP\_FETCH\_INTRST$ is found to be 94.86%. By using the calculated cumulative footprints in Table 2 for full inflection zones to the left of $X_{int\,rst}^*$, by beginning numerical integration in the zone in which $X_{int\,rst}^*$ is located, and by only performing integration up to the value of $X_{int\,rst}^*$, the number of iterations is confined to be no greater than $n$ (Appendix C).



### 3.3    *FETCH*_70, *FETCH*_80, and *FETCH*_90

*FETCH*_$p$, where suffix $p$ can be 70, 80, or 90, is the conversion of the corresponding nondimensional form $X_p^*$ to its field scale, or dimensional form, using Eq. (13). Therefore, similarly to the derivation of Eq. (21), the conversion of *FETCH*_$p$ from $X_p^*$ is given by:

$$FETCH\_p = kX_p^* z \frac{\overline{u}(z)}{u_*}\left(1-\frac{z}{h}\right)^{-1}.$$
(29)

Since the values of $k$, $z$, $\overline{u}(z)$, $u_*$, and $h$ can be acquired in the same way as for Eq. (21), $X_p^*$ is additionally needed. Whereas $X_p^*$ is the nondimensional upwind fetch within which the horizontal uniform sources of gas flux contribute $p$% of the measured flux, it is mathematically expressed as:

$$100\int_{X_S^*}^{X_p^*} F_y^*\left(X^*\right)dX^* = p.$$
(30)

The data in Table 2 indicate $X_p^* > X_{F2}^*$ while $p \geq 32.53\%$. For $p$ equal to 70, 80, or 90, the left side of Eq. (30) can therefore be expressed in two terms:

$$100\int_{X_S^*}^{p} F_y^*\left(X^*\right)dX^* = 100\int_{X_S^*}^{X_{F2}^*} F_y^*\left(X^*\right)dX^* + 100\int_{X_{F2}^*}^{X_p^*} F_y^*\left(X^*\right)dX^*,$$
(31)

where the 1st term on the right side of this equation is a constant, given in Table 2 for the two ranges of boundary-layer stabilities. If this constant is denoted by $P_{X_{F2}^*}$, the range over which to integrate can be made smaller, beginning at $X_{F2}^*$, 

instead of $X_S^*$, and extending to $X_p^*$:

$$p_{X_{F2}^*} + 100\int_{X_{F2}^*}^{X_p^*} F_y^*\left(X^*\right)dX^* = p.$$
(32)

In the integration term of this equation, 25 may be used as an upper limit for $X^*$ because 90% of fluxes will always be below 25, which is also why Fig. 2 of Kljun et al. (2015) only extends to 25. Thus, an increment in $X^*$ can be evaluated by

$$\Delta X^* = \frac{25 - X_{F2}^*}{1000}$$
(33)

To find $X_p^*$ from Eq. (32), $X_p^*$ needs to be expressed explicitly.





Although an integer $m$ rarely exists that satisfies $X_{F2}^* + m\Delta X^* = X_p^*$, it can easily hold that $X_{F2}^* + m\Delta X^* \approx X_p^*$, from which an explicit equation for $X_p^*$ can be derived. In this case, the inequality $X_{F2}^* + m\Delta X^* > X_p^*$ can lead to:

$$p_{X_{F2}^*} + 100 \int_{X_{F2}^*}^{X_{F2}^* + m\Delta X^*} F_y^*(X^*) dX^* = p_+ \geq p, \tag{34}$$

where $m$ can be between 1 and 1000 as long as $X^* < 25$ (Eq. 33). Since Eq. (34) integrates to an upper limit that is slightly greater than $X_p^*$ and the result is slightly greater than $p$, we should also find the upper limit and result that is barely less than $X_p^*$ and $p$, respectively. This limit must be $X_{F2}^* + (m-1)\Delta X^* < X_p^*$, which yields:

$$p_{X_{F2}^*} + 100 \int_{X_{F2}^*}^{X_{F2}^* + (m-1)\Delta X^*} F_y^*(X^*) dX^* = p_- \leq p. \tag{35}$$

Now $X_p^*$ is a value bounded by $X_{F2}^* + (m-1)\Delta X^*$ and $X_{F2}^* + m\Delta X^*$. In the process of numerical integration, the values of $p_+$, $p_-$, and $m$ can be easily identified (Appendix C). The following section shows how Eqs. (32) to (35) may be used to find a solution to $X_p^*$.

### 3.3.1 Solution to $X_p^*$

Equation (34) minus (32) leads to:

$$100 \int_{X_p^*}^{X_{F2}^* + m\Delta X^*} F_y^*(X^*) dX^* = p_+ - p. \tag{36}$$

The Intermediate Value Theorem reforms this equation as

$$100 \left( X_{F2}^* + m\Delta X^* - X_p^* \right) F_y^*(X_\xi^*) = p_+ - p, \tag{37}$$

where $X_\xi^*$ is an intermediate value in the range from $X_p^*$ to $X_{F2}^* + m\Delta X^*$ and makes $F_y^*(X_\xi^*)$ equal to the average of $F_y^*(X^*)$ over the range. Similarly, Eq. (32) minus (35) leads to:

$$100 \left[ X_p^* - X_{F2}^* - (m-1)\Delta X^* \right] F_y^*(X_\zeta^*) = p - p_-, \tag{38}$$

where $X_\zeta^*$ is an intermediate value in the range from $X_{F2}^* + (m-1)\Delta X^*$ to $X_p^*$ and makes $F_y^*(X_\zeta^*)$ equal to the average of $F_y^*(X^*)$ over this range. Because both $X_\xi^*$ and $X_\zeta^*$ are very close, in fact within a range as small as the size of $\Delta X^*$, and whereas $F_y^*(X^*)$ is a continuous function and both $F_y^*(X_\xi^*)$ and $F_y^*(X_\zeta^*)$ can be considered almost equal, their ratio tends to be 1. As a result, the ratio of Eq. (37) to (38) leads to:



$$\frac{X_{F2}^{*} + m\Delta X^{*} - X_{p}^{*}}{X_{p}^{*} - X_{F2}^{*} - (m-1)\Delta X^{*}} \approx \frac{p_{+} - p}{p - p_{-}} . \tag{39}$$

If this equation is solved for $X_{p}^{*}$, the result is an interpolation equation:

$$X_{p}^{*} \approx X_{F2}^{*} + \left( m - \frac{p_{+} - p}{p_{+} - p_{-}} \right) \Delta X^{*} \tag{40}$$

Now $X_{p}^{*}$ may be calculated, and its result used in Eq. (29) to calculate *FETCH_p*.

### 3.3.2    Example

In order to acquire *FETCH*_70 for the same conditions as described in section 3.2.3, we use numerical integration as shown in Eqs. (34) and (35) (see Appendix C for application of integration) to find the inputs needed for the interpolation

in Eq. (40), which results in $X_{70}^{*}$ (i.e., $X_{p}^{*}$ at subscript $p = 70$). Given the value of $X_{F2}^{*}$ from Table 1, $\Delta X^{*}$ can be computed

from Eq. (33) to be $2.3662560 \times 10^{-2}$.  At $m = 102$, $p_{+} = 70.114313$ from Eq. (34), and $p_{-} = 69.868805$ from Eq. (35). Next,

$X_{70}^{*}$ can be computed from Eq. (40) as

$$X_{70}^{*} \approx X_{F2}^{*} + \left( m - \frac{p_{+} - 70}{p_{+} - p_{-}} \right) \Delta X^{*} = 3.7400033 \tag{41}$$

Using this value, Eq. (29) generates the following result:

$$FETCH\_70 = kX_{70}^{*} z \frac{\overline{u}(z)}{u_{*}} \left( 1 - \frac{z}{h} \right)^{-1} = 102.65 \, \text{m} \tag{42}$$

This example illustrates that instead of extensive numerical integrations in the field, Eqs. (34) and (35) may be solved beforehand for $X_{70}^{*}, X_{80}^{*},$ and $X_{70}^{*}$ (Table 3) due to $X_{p}^{*}$ being independent of field measurements. Then, these values, along with field measurements, may be used in Eq. (29) to find their final field scale values.

**Table** 3 Nondimensional upwind fetch $X_{p}^{*}$, where subscript $p$ indicates 70, 80, or 90.  At a nondimensional scale, a $p\%$ portion of the

measured flux is contributed by its footprint area within $X_{p}^{*}$, assuming the sources/sinks of passive gas are uniform over a flat field. ($z$ is aerodynamic height for measurements and $L$ is Monin-Obukhov length.)

| Atmospheric stability | $X_{70}^{*}$ | $X_{80}^{*}$ | $X_{90}^{*}$ |
|---|---|---|---|
| Convective $z/L < 0$ | 3.7400033[a] | 5.5734341 | 10.371083 |
| Neutral/stable $z/L \geq 0$ | 4.3702906 | 6.9142010 | 14.612024 |

[a] For each number, at least eight digits are kept for significance of computations in single precision.



## 4    Discussion

This study details the application of Kljun et al's. (2015) flux footprint equations (Eq. 19) into eddy-covariance flux
systems so that footprint characteristics of measured flux can be computed every interval of flux data output in the field.
These computed flux footprint characteristics are those required by datasets documented in AmeriFlux (2018) and adopted
by regional, national, and international flux networks (e.g., NYS Mesonet, ChinaFlux, and FluxNet). Previously, these
characteristics have been evaluated only through computationally laborious numerical integration (Kormann and Meixner,
2001; Kljun et al., 2002; 2025), not suitable for the limited computation capacity typically found in field computer
processors. Therefore, the development in this study focuses on field computation-saving methodologies, now adopted into
the EasyFlux series programs (Campbell Scientific Inc, 2022). Indeed, the nondimensional forms of fetch (Eq. 13) and
footprint equations (Eq. 19) from Eqs. (6) to (14) in Kljun et al. (2015) make field computation-saving methodologies
applicable (Appendix C).

It should be noted that the naming and selected footprint variables in this study were chosen to be in conformity with the
2018 AmeriFlux data variable format.  Furthermore, data precision was optimized to match the computation precision inside
field eddy-covariance flux systems.  And lastly, the algorithm for the estimation of planetary boundary layer height $h$ from
measured variables in an eddy-covariance flux system was a major consideration for this study, and details concerning it are
described in Appendix B. Beyond the immediate applications in this study, the developed equations found herein and in
Kljun et al. (2015) have important implications for the optimization of eddy-covariance measurement height in order to
maximize the proportion of measured flux from the footprint area of most interest. In the following sections, more discussion
is given to the merits of Eq. (19), the expression of variables, the optimization of data precision, the algorithm for $h$, and
more applications of equations.

### 4.1    Merits of Kljun et al's. (2015) flux footprint equations

Computing flux footprint characteristics such as *FETCH_p*, where subscript $p$ is 70, 80, or 90, and *FP_FETCH_INTRST*,
has typically been challenging in the field because approaches like Hsieh et al (2000) or Kormann and Meixner (2001)
require computationally laborious numerical integrations.  The use of nondimensional flux footprint equations found in
Kljun et al. (2015) can reduce or fully avoid numerical integration. For *FETCH_p*, given Table 3, only an analytical equation
(Eq. 29) is needed, requiring a simple algebraic calculation. For *FP_FETCH_INTRST,* given Table 2, Eqs. (22), and (26), the
numerical integration is reduced to a fractional zone, as shown in Fig 2, from one turning point to $X_{\text{intrst}}^{*}$ .

### 4.2    Variable expressions

The names of some variables in this study, such as *FP_FETCH_INTRST*, *FETCH_MAX*, *FETCH_70*, *FETCH_80*, and
*FETCH_90* and *FP_FETCH_INTRST*, are lengthy but used for the sake of consistency with the data variable format
documented in AmeriFlux (2018).

### 4.3    Data precision





Unlike a desktop or laptop computer, a computation module like a CR6 or CR1000X datalogger is smaller in size, lower in power consumption, and more durable in rugged environment conditions, plus it has multiple functionalities for control, measurement, communication, computation, and data storage. As such, the performance of a microprocessor inside the computation module is optimized for all mentioned functionalities through balancing its size, power consumption, and durability. For optimization, single precision is used for data processing inside the microprocessor. Accordingly, eight

significant digits in single precision are kept for the data shown in the three data tables and Eqs. (27), (28), and (41) of this paper. However, it should be noted that these data were computed from Eq (19) using double precision processing on a desktop computer, even though the precision of data from Eq. (19) is hardly warranted because it depends on the precision of equation parameters that were statistically estimated (section 2.5). Nonetheless, considering Eq. (19) as an exact equation, this study carefully warrants the accuracy of numerical integrations and the precision of data for computations of flux

footprint characteristics.

### 4.4 Algorithm for planetary boundary layer height

      The planetary boundary-layer height ($h$) is one of the required variables in the flux footprint equations of Kljun et al. (2015), (see Eqs. 21, 22, 29, and 42). Unlike other variables, it is not directly measured or commonly computed from measured data in eddy-covariance systems. And while it can be directly measured using a ceilometer (e.g., SkyVUE Pro,

Campbell Scientific Inc. 2025), it is often cost prohibitive. As shown in Eqs. (B1) and (B3), $h$ is theoretically related to other commonly measured variables in an eddy-covariance system. Since the main body of this paper focuses on computations of flux footprint characteristics, this algorithm is developed in Appendix B, although the algorithm is still a key finding from this study.

### 4.5 Applications of equations developed in this study

Equation (33) is used to calculate a $\Delta X^*$ value for use in Eqs. (34) to (40). In reference to Fig. 2 of Kljun et al. (2015), an assumed top limit of 25 for $X^*$ is used for this calculation. Between $X_S^*$ and 25, the integration of Eq. (19) is equal to 96.50% and 93.98% for convective and neutral/stable atmospheric stabilities, respectively. Accordingly, in Eqs. (34) to (40), the $p$ value should be $\leq$ 96.50% under convective atmospheric stability or $\leq$ 93.98% under neutral/stable atmospheric stability. In the case that $p$ is above these ranges, the value from Eq. (33) is still applicable because it has a higher resolution than if 25

were replaced with a higher value in Eq. (33). Alternatively, $\Delta X^*$ also can be extended or narrowed, depending on the accuracy required for *FETCH_p*.

      Although Eq. (40) was developed for cases of $p$ equal to 70, 80, or 90 to compute *FETCH*_70, *FETCH* _80, or *FETCH* _90, it can be used for any $p$ value. In reference to the cumulative footprint values in Table 2, $X_{F2}^*$ in this equation can be replaced with $X_{F1}^*$, $X_{max}^*$, or $X_S^*$, depending on the $p$ value under different atmospheric stabilities. In reference to the

integration resolution values also in Table 2, the integration resolution for the corresponding zone can be used as a value of $\Delta X^*$.




For example (Table 2), $X_{F2}^{*}$ in Eq. (40) should be replaced with $X_{S}^{*}$ under convective atmospheric stability if $p <$ 1.1321783 or under neutral/stable atmospheric stability if $p < 0.87452260$, in which case $\Delta X^{*}$ would be $4.1726699 \times 10^{-4}$ and $3.1310199 \times 10^{-4}$, respectively.

### 4.6 Optimize measurement height

Perhaps the most significant application of flux footprint equations is the optimization of measurement height of eddy-covariance sensors (i.e., $z_{\mathrm{m}}$, the height of measurement volume center above the ground), such as a sonic anemometer and a gas analyzer (Horst and Weil, 1994). Over a flat field with uniform flux sources/sinks, the higher the measurement volume, the farther the flux footprint can extend away from the flux tower whereas the lower the measurement volume, the closer the flux footprint converges to the flux tower (Fig. 1). Over a flat field, $z_{\mathrm{m}}$ is generally optimized for an expected percentage $p$ of measured flux from a given upwind fetch or for maximization of measured flux contribution from a targeted area covered by an ecosystem of interest.

#### 4.6.1 Optimization of $z_{\mathrm{m}}$ for an expected percentage of measured flux within a given upwind fetch

Given an upwind fetch, $FETCH\_p$, from which a $p\%$ measured flux is expected, Eq. (29) can be rearranged and solved for the optimized height, $z_{\mathrm{mp}}$:

$$z_{mp} = \frac{u_{*}h FETCH\_p}{kh\overline{u}\left(z\right)X_{p}^{*} + u_{*}FETCH\_p} + d \ , \tag{43}$$

where $z_{\mathrm{mp}}$ is chosen for the prevailing atmospheric stability at a site (e.g., the case in section 3.2.3) since the height of system sensors is typically inconvenient to adjust after installation. As an example, given $FETCH\_90$ to be 500 m, atmospheric stability as described in section 3.2.3, $d$ to be 0.25 m, and $X_{90}^{*}$ from Table 3 for $z/L < 0$, Eq. (43) generates $z_{\mathrm{mp}}$ to be 9.00 m.

Equation (43) describes $z_{\mathrm{mp}}$ essentially as a function of $FETCH\_p$ because the other aerodynamic variables in the equation are given for a site's prevailing atmospheric stability. Using $X_{70}^{*}, X_{80}^{*},$ and $X_{90}^{*}$ values from Table 3, $z_{\mathrm{m}70}, z_{\mathrm{m}80}$, and $z_{\mathrm{m}90}$ corresponding to $FETCH\_70$, $FETCH\_80$, and $FETCH\_90$ under the prevailing atmospheric stability can be generated from Eq. (43). For any percentage of measured gas flux from a given upwind fetch $FETCH\_p$, $X_{p}^{*}$ value needed by Eq. (43) can be numerically computed from Eq. (40).

#### 4.6.2 Optimization of $z_{\mathrm{m}}$ to maximize measured flux from the targeted area of an ecosystem of interest

A common practice in eddy-covariance system installation is to optimize $z_{\mathrm{m}}$. The optimization aims to maximize the measured fluxes from the targeted area covered by an ecosystem of interest while minimizing the influence of fluxes from the area covered by undesirable ecosystems outside the target area and from the fenced area disturbed by station facilities (e.g., supporting structure), instruments for other micrometeorological variables (e.g., radiation, soil moisture, and rain), and solar panels for power supply to the system (Fig. 3). The degree of influence depends on many factors such as the type and area of undesirable ecosystems, the size of fenced areas, the volume of facilities, the surface of solar panels. Although the fluxes from the undesirable ecosystems and the disturbed area will unavoidably contaminate the measurement




volume, it can be minimized through the optimization of $z_m$ (Kormann and Meixner, 2001). Depending on surface roughness mostly accounted by $d$ along with $\overline{u}(z)$ gradient and atmospheric boundary-layer stability accounted by $h$ (Rebmann et al., 2018), for the optimization of $z_m$, the fraction of measured flux from the targeted area can be evaluated from flux footprint equations.



**Figure 3**: A drone view of field situation in a case that a closed-path eddy-covariance system (i.e., CPEC310, Campbell Scientific, Inc., UT, USA) was used to measure the $CO_2$ and $H_2O$ fluxes over *Haloxylon ammodendron* plantation near bare sand land (farther top area) in Minqin, China. As view, the installation height of CPEC310 sensors should be optimized to maximize the measured fluxes from the area inside the external and outside the inner circles while minimizing the measured fluxes from both the bare sand land area outside the external circle and the fenced area covered by flux tower, weather station, solar panel, ceilometer (SkyVue), and instrument for soil moisture and soil temperature. This view is not scaled.

The targeted area is generally in the shape of an annulus centered at the flux tower (Fig. 3) with its external radius $R$, outside which the area is covered by undesirable ecosystems, and with its inner radius $r$, inside of which a fenced portion is



the disturbed area. The optimization of $z_m$ is to find a height at which the portion of measured flux from the annulus footprint area is maximized. This portion denoted by $F_{pa}$, where subscript $a$ indicates annulus, is given from Eq. (19) as

$$F_{pa} = \int_{X_r^*}^{X_R^*} F_y^*(X^*)dX^* \tag{44}$$

where $X_r^*$ is the nondimensional fetch corresponding to the inner annulus radius $r$ at field scale and is given from Eq. (13) as

$$X_r^* = \frac{r}{z}\left(1 - \frac{z}{h}\right)\left(k\frac{u(z)}{u_*}\right)^{-1} \tag{45}$$

and $X_R^*$ is the nondimensional fetch corresponding to the outer annulus radius $R$ at field scale, given also from Eq. (13) as

$$X_R^* = \frac{R}{z}\left(1 - \frac{z}{h}\right)\left(k\frac{u(z)}{u_*}\right)^{-1}. \tag{46}$$

Given $r$ and $R$ under a specified boundary-layer condition, both $X_r^*$ and $X_R^*$ change with $z$. For a prevailing boundary layer condition with given $h$, $u_*$, and $u(z)$, $F_{pa}$ is a function of $z$ [i.e., $F_{pa}(z)$], with the integration limits of Eq. (44) varying with $z$. The $z$ value at which $F_{pa}(z)$ reaches its maximum is the optimum aerodynamic height, denoted by $z_{\max}$. This height is the solution to

$$\left.\frac{dF_{pa}(z)}{dz}\right|_{z=z_{\max}} = 0 \tag{47}$$

At $z_{\max}$, the measurement volume of an eddy-covariance system will receive the largest possible portion of fluxes from the annulus area of interest. For the solution of $z_{\max}$, we find the derivative of $F_p(z)$ with respect to $z$:

$$\frac{dF_{pa}(z)}{dz} = -\frac{u_*}{ku(z)z^2}\left[RF_y^*(X_R^*) - rF_y^*(X_r^*)\right] \tag{48}$$

Given $r$ and $R$ values, $X_r^*$ and $X_R^*$ can be computed from Eqs. (45) and (46), respectively. In reference to Eq. (19), an analytical solution to $z_{\max}$ for Eq. (47) from Eq. (48) is unavailable, but it can be found graphically, as shown in Fig. 4 for a case of $r = 15$ m and $R = 300$ under the same boundary-layer conditions as in section 3.2.3. The result is accurate to within a centimeter, plenty for sensor installation. In Fig. 4, the $dF_{pa}(z)/dz$ curve crosses the $z$-axis at $z_{\max}$, which is 5.71 m. At $z_{\max}$, $F_{pa}(z)$ exactly reaches its maximum of 84.42% (see dashed line in Fig. 4). Given that $d$ is 0.25 m, $z_m$ can be optimized as 5.96 m (i.e., $z_m = z_{\max} + d$). This optimization methodology was developed by the authors while specifying installation of eddy-covariance sensors at Moorefield, Wellfleet, and Benkelman in Nebraska, USA.





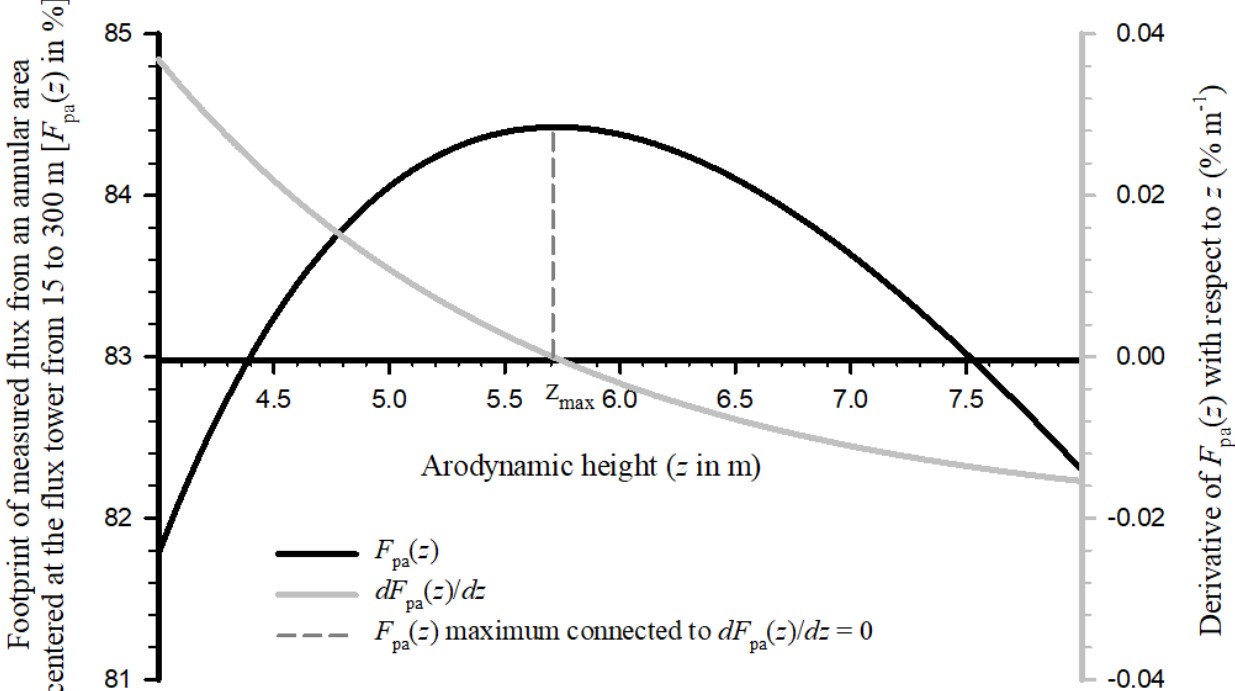

**Figure** 4: Graphical optimization of the aerodynamic height of the eddy-covariance measurement volume $z_{max}$ to maximize the portion of measured flux from the annular area centered at the flux tower (e.g., from its inner radius 15 to its external radius 300 m). $F_{pa}(z)$ values are computed from Eq. (19) for $z/L < 0$, where $L$ is Monin-Obukhov length, and from Eqs. (44) to (46). Values for $dF_{pa}(z)/dz$ are computed from Eq (48). The optimized aerodynamic height $z_{max}$ is 5.71 m when $F_{pa}(z)$ reaches its maximum of 84.42% and where its derivative with respect to $z$ is zero (see dashed line). Given the zero-plane displacement height $d$ to be 0.25 m, the measurement height $z_m$ can be optimized as 5.96 m (i.e., $z_m = z_{max} + d$). Wind speed is 4.00 m s$^{-1}$, friction velocity is 0.30 ms$^{-1}$, and planetary boundary layer height is 1200 m.

## 5    Summary remarks

Flux datasets are increasingly requiring the inclusion of flux footprint fetch characteristics, specifically the upwind maximum footprint location and the upwind percentage fetches (AmeriFlux, 2018). An additional flux footprint fetch characteristic included in many ChinaFlux datasets is the percentage of measured fluxes attributable to an area of interest. In a field eddy-covariance flux system, these characteristics are ideally evaluated simultaneously with the computations of the flux data every interval of these data output (e.g., 30 min). In order for such evaluations to be time-efficient inside the microprocessor of a field datalogger, time-saving algorithms that retain accuracy through every step are developed from the well-accepted flux footprint equations of Kljun et al. (2015) (i.e., Eq. 19).

As a merit of Kljun et al. (2015), the upwind maximum footprint location, inflection locations, and upwind percentage fetches from their flux footprint equations are in the nondimensional domain, are invariant (Fig. 2, Tables 1 and 3), and can be precisely computed beforehand in a laboratory. Similarly, by using analytical Eqs. (21) and (29), the maximum footprint location and upwind percentage fetches can be converted from their non-dimensional data in Table 3 to field scale units and



then stored inside the microprocessor for immediate use (Appendix C), thus avoiding the use of numerical integration in the field. And finally, because of this merit, the data in Table 3 also reduces the computation load for the interest footprint to a limited amount less than an inflection zone (Eq. 27).

The accuracy of the computed footprint characteristics is considered through the division of the footprint equation curve into four inflection zones for integration (Fig. 2, Tables 1 and 2). According to the comments in Appendix A of Kljun et al. (2015), better accuracy in the flux footprint characteristics leads to us adopting the equations with parameters from their Table A1 for convection and neutral/stable atmospheric boundary layer stabilities (Eq. 19, Fig. 2), instead of using their universal flux footprint Eqs. (14) and (17) of Kljun et al. (2015). Where possible, eight significant digits of data (Tables 1, 2,

and 3; Eqs. 27, 28, and 41) are kept for all computations at single precision, which is an additional consideration to warrant the accuracy in the flux footprint characteristics.

As shown in all application equations in this study (e.g., Eq. 21), the planetary boundary-layer height is needed as a scaling variable for flux footprint equations of Kljun et al. (2015), but it is not commonly directly measured with field eddy-covariance systems. For this variable to be acquired every data output interval from other variables measured by eddy-

covariance flux systems in the field, the applicable algorithm is developed in Appendix B.

As shown in Model (15), nondimensional upwind fetch ($X^*$) is the independent variable of flux footprint equations. An explicit expression for this fetch or for nondimensional upwind percentage fetch is not available. Thus, a numerical equation for nondimensional upwind percentage fetch is theoretically derived (Eqs. 29 to 40) and a conversion into field scale is shown.

Our discussions go beyond the focus of this study for the most practical and significant application of flux footprint equations in eddy-covariance flux measurements, that is to optimize the installation height of eddy-covariance sensors. Optimization means 1) to ensure an expected percentage of measured flux from a targeted upwind fetch and 2) to maximize the contribution of measured fluxes from the footprint area of interest. The methodology for this optimization is additionally discussed (Figs. 3 and 4, Eqs. 43 to 48). With this addition, this study more fully documents the common applications of

Kljun et al. (2015) to field eddy-covariance flux systems. This document is intended to be a reference source for flux footprint equation applications, especially for users and developers of EasyFlux series programs found in many Campbell Scientific eddy-covariance flux systems globally.

**Code and data availability:**

The program code related to the methods and algorithms that were developed in this manuscript is available from
https://doi.org/10.5281/zenodo.18143076 under (CC-BY-4.0) license, as are input data to produce the plots for all the simulations presented in this paper (Zhou and Chen, 2025).

**Acknowledgements:**

This study is a collaboration effort of Ker Laboratory under Qingyuan Forest CERN with ChinaFlux. It is supported by the Global Science Program of Campbell Scientific Inc. Authors thank Dr. Prajaya Prajapati for his review, Mr. Bart
Ransbottom for his figure art, and Dr. Dirk Baker for his coordination





**Appendix A: Maximum and inflection locations on the crosswind-integrated footprint curve**

At the maximum of the nondimensional crosswind-integrated footprint $F_y^*(X^*)$, its 1$^{st}$ order derivative with respect to nondimensional upwind fetch $X^*$ should satisfy

$$\left. \frac{dF_y^*(X^*)}{dX^*} \right|_{X^*=X_{max}^*} = 0, \tag{A1}$$

where $X_{max}^*$ is its maximum location. From Model (15), this 1$^{st}$ order derivative is given by

$$\frac{dF_y^*(X^*)}{dX^*} = a \left[ b\left(X^*-d_0\right)^{b-1} + c\left(X^*-d_0\right)^{b-2} \right] \exp\left( -\frac{c}{X^*-d_0} \right). \tag{A2}$$

Following Eq. (A1), setting this equal to zero leads to:

$$b\left(X_{max}^*-d_0\right)^{b-1} + c\left(X_{max}^*-d_0\right)^{b-2} = 0, \tag{A3}$$

and solving for $X_{max}^*$ leads to Eq. (20).

At an inflection point of $F_y^*(X^*)$, its 2$^{nd}$ order derivative with respect to $X^*$ must satisfy

$$\left. \frac{d}{dX^*}\left( \frac{dF_y^*(X^*)}{dX^*} \right) \right|_{X^*=X_I^*} = 0, \tag{A4}$$

where $X_I^*$ is the nondimensional upwind inflection location on the curve of $F_y^*(X^*)$ and its subscript I indicates inflection. This subscript can be F1 or F2 for the 1$^{st}$ or 2$^{nd}$ inflection locations (Fig. 2). Equation (A4) is a further derivative of Eq. (A2), given by:

$$\frac{d}{dX^*}\left( \frac{dF_y^*(X^*)}{dX^*} \right) = a \exp\left( -\frac{c}{X^*-d_0} \right) \left[ b(b-1)\left(X^*-d_0\right)^{b-2} + 2c(b-1)\left(X^*-d_0\right)^{b-3} + c^2\left(X^*-d_0\right)^{b-4} \right]. \tag{A5}$$

To satisfy Eq. (A4),

$$b\left(X_I^*-d_0\right)^2 + 2c\left(X_I^*-d_0\right) + \frac{c^2}{b-1} = 0. \tag{A6}$$

The solutions to $X_I^*$ from this equation are the two inflection locations in terms of nondimensional upwind fetch $X_{F1}^*$ and $X_{F1}^*$ given in Eqs. (24) and (25), respectively, and shown in Fig. 2.





## Appendix B: Estimation of planetary boundary layer height from measured variables in eddy-covariance flux systems

In order to compute the footprint characteristics using equations of Kljun et al. (2015), the planetary boundary layer height ($h$) is required by Eqs. (21), (22), (29), (42), (45), and (46). Fortunately, it may be estimated using commonly measured variables in eddy-covariance systems. Appendix B in Kljun et al. (2015) summarizes the equations for $h$ under different atmospheric boundary-layer stratifications and recommends theoretical equations of $h$ for use in eddy-covariance flux systems.

### B1 Equations of $h$ for use in eddy-covariance flux systems

For neutral to stable conditions, Kljun et al. (2015) summarized four equations. One is the primary equation, while the other three are the simplified versions for extreme cases of free atmosphere or strongly stable boundary-layer conditions. The primary equation is an interpolation formula proposed by Nieuwstadt (1981):

$$h = \frac{L}{3.8}\left( \sqrt{1 + 2.28\frac{u_*}{fL}} - 1 \right), \tag{B1}$$

where $L$ is Monin-Obukhov length, $u_*$ is friction velocity, and $f$ is the Coriolis parameter. In eddy-covariance flux systems, mean values of $L$ and $u_*$ (Rebmann et al., 2012) are computed every data output interval (e.g., 30 min), while $f$ can be computed at any time from (Wallace and Hobbs, 2006)

$$f = 2\Omega \sin \phi , \tag{B2}$$

where $\Omega$ is the angular velocity of Earth's rotation ($7.2924621e^{-5}$ rad s$^{-1}$) and $\phi$ is the latitude of an eddy-covariance flux station. As a station variable, $\phi$ is entered by a user into an eddy-covariance flux system before or while an EasyFlux series program is running.

For convective atmospheric conditions, an equation explicit to $h$ is not available, however its differential equation with respect to time $t$ is defined by Batchvarova and Cryning (1991) as

$$\frac{dh}{dt}\left( \frac{\gamma h^2}{(1+2A)h - 2BkL} + \frac{Cu_*^2 T}{g(1+A)h - gBkL} \right) = \overline{w'\Theta'}_0 , \tag{B3}$$

where $\gamma$ is dry adiabatic lapse rate (commonly $9.8\times10^{-3}$ K m$^{-1}$), $A$, $B$, and $C$ are parameters, $k$ is the von Karman constant (0.41), $g$ is acceleration due to gravity (9.81 m s$^{-2}$ at sea level), $w'$ is vertical wind fluctuation, $\Theta'$ is potential air temperature fluctuation, and $\overline{w'\Theta'}_0$ is the covariance of $w$ with $\Theta$, which drives the sensible heat flux over the interface between ecosystems and the atmosphere. Over this interface, $\overline{w'\Theta'}_0$ can be substituted with the covariance of $w$ with air temperature



($T$), denoted by $\overline{w'T'}$, where $T'$ is air temperature fluctuation. This covariance is available in eddy-covariance flux systems. At present, an exact solution to $h$ from Eq. (B3) is not available, but a numerical solution may be expressed as a divided difference form.

**B2 The divided difference form of $h$ terms in Eq. (B3)**

In eddy-covariance flux systems, the aerodynamic and thermodynamic variables used for Eq. (B3), such as $u_*$, $L$, $T$, and $\overline{w'T'}$, are computed from measured data averaged over a data output interval, denoted by $\Delta t$. As such, $h$ can be derived only on a temporal scale of $\Delta t$. Given $h_b$ to be a $h$ value at the beginning of $\Delta t$, and $h_e$ at the end, the derivative term can be expressed as

$$\frac{dh}{dt} = \frac{h_e - h_b}{\Delta t}, \tag{B4}$$

where, under continuous measurements, $h_b$ over current $\Delta t$ is $h_e$ over a previous one. While the boundary layer is developing, $h_b$ and $h_e$ are rarely equal, and over a short period of $\Delta t$ the change from $h_b$ to $h_e$ can be reasonably assumed to be linear. Accordingly, a $h$ value can be approximated from

$$h = \frac{h_e + h_b}{2}, \tag{B5}$$

Apparently, $h$ value can be acquired if $h_e$ value is estimated at the end of current $\Delta t$. In Eq. (B3), substitution of $dh/dt$ and $h$ with their corresponding divided difference forms (i.e., Eqs. B4 and B5) leads to

$$\frac{\gamma(1+A)}{8\Delta t}h_e^4 + \frac{\gamma}{4\Delta t}\left((1+A)h_b - BkL\right)h_e^3 - \left\{\frac{\gamma BkL}{4\Delta t}h_b - \frac{1+2A}{2}\left[\frac{Cu_*^2 T}{\Delta t g} - (1+A)\frac{\overline{w'T'}}{2}\right]\right\}h_e^2$$

$$-\left\{\frac{\gamma(1+A)}{4\Delta t}h_b^3 - \frac{\gamma BkL}{4\Delta t}h_b^2 + \frac{(1+A)(1+2A)}{2}\overline{w'T'}_0 h_b - BkL\left[(3+4A)\frac{\overline{w'T'}}{2} - \frac{2Cu_*^2 T}{\Delta t g}\right]\right\}h_e \tag{B6}$$

$$= \frac{\gamma(1+A)}{8\Delta t}h_b^4 - \frac{\gamma BkL}{4\Delta t}h_b^3 + \frac{1+2A}{2}\left[\frac{Cu_*^2 T}{\Delta t g} + (1+A)\frac{\overline{w'T'}}{2}\right]h_b^2 - BkL\left[(3+4A)\frac{\overline{w'T'}}{2} + \frac{2Cu_*^2 T}{\Delta t g}\right]h_b + 2(BkL)^2 \overline{w'T'}.$$

After the parameters $A$, $B$, and $C$ in this equation are replaced with their corresponding values 0.2, 2.5, and 8.0 from Appendix B in Kljun et al. (2015), the equation becomes





$$0 = 0.15 \frac{\gamma}{\Delta t} h_e^4 + \frac{\gamma}{\Delta t}(0.3h_b - 0.625kL)h_e^3 - \left(0.625\frac{\gamma kL}{\Delta t}h_b - 5.6\frac{u_*^2 T}{\Delta t g} + 0.42\overline{w'T'}\right)h_e^2$$

$$-\left[0.3\frac{\gamma}{\Delta t}h_b^3 - 0.625\frac{\gamma kL}{\Delta t}h_b^2 + 0.84\overline{w'T'}h_b + kL\left(40.0\frac{u_*^2 T}{\Delta t g} - 4.75\overline{w'T'}\right)\right]h_e$$

(B7)

$$-0.15\frac{\gamma}{\Delta t}h_b^4 + 0.625\frac{\gamma kL}{\Delta t}h_b^3 - \left(5.6\frac{u_*^2 T}{\Delta t g} + 0.42\overline{w'T'}\right)h_b^2 + kL\left(40.0\frac{u_*^2 T}{\Delta t g} + 4.75\overline{w'T'}\right)h_b - 12.5k^2L^2\overline{w'T'}.$$

Inside this equation, the only unknown variable is $h_e$, and since the equation is its 4th order polynomial, there are four possible solutions. One of the positive root values $h_b \in [h_b \pm \delta]$, where $\delta > 0$, must be the solution to $h_e$. Unfortunately, an explicit solution to $h_e$ from this equation is not available, so a numerical method must be used.

**B3 Numerical solution to $h_e$**

If $h_e$ on the right side of Eq. (B7) is replaced with $h_x$ and represents a value $h_b \in [h_b \pm \delta]$, and 0 on the left side is replaced with $f(h_x)$ but still equals zero in the case of $h_x = h_e$, then $f(h_x)$ is a continuous differentiable function with a non-zero 1st order derivative. Therefore, the Newton-Raphson numerical method is applicable to the solution of $f(h_x)$ at zero for $h_e$ (Burden et. al., 2016).

Suppose that $f(h_e) \in C^4[h \pm \delta]$ and let $h_b \in [h_b \pm \delta]$ be an initial approximation to $h_e$ value such that $f'(h_x)\big|_{h_x = h_0} \neq 0$ and $|h_e - h_b|$ is sufficiently "small". Then, the 2nd order Taylor polynomial for $f(h_x)$ about $h_b$ is given by:

$$f(h_x) = f(h_b) + (h_x - h_b)f'(h_b) + \frac{(h_x - h_b)^2}{2}f''(h_\xi),$$

(B8)

where $h_\xi$ lies between $h_b$ and $h_x$. Since $f(h_e) = 0$, this equation becomes

$$f(h_b) + (h_e - h_b)f'(h_b) + \frac{(h_e - h_b)^2}{2}f''(h_\xi) = 0.$$

(B9)

Because $h_\xi$ is unknown, $h_e$ cannot be resolved from this equation, but after the 2nd order term with $f''(\xi)$ is dropped, Eq. (B9) is commonly written as

$$h_e \approx h_b - \frac{f(h_b)}{f'(h_b)}.$$

(B10)

The right side of this equation is the $h_x$-intercept of the tangent line of $f(h_x)$ at $[h_b, f(h_b)]$. The value of this intercept can be denoted by $h_{e_1}$ and is a first approximation for $h_e$. In Eq. (B10), the approximation sign can become an equal sign if $h_{e_1}$ is used to replace $h_e$:




$$h_{e_1} = h_b - \frac{f(h_b)}{f'(h_b)},$$
(B11)

where

$$f(h_b) = \left(-1.68h_b^2 + 9.5kLh_b - 12.5k^2L^2\right)\overline{w'T'},$$
(B12)

and

$$f'(h_b) = 1.20\frac{\gamma}{\Delta t}h_b^3 - 2.50\frac{\gamma kL}{\Delta t}h_b^2$$
$$+ \left(11.2\frac{u_*^2T}{\Delta t g} - 1.68\overline{w'T'}\right)h_b - kL\left(40.0\frac{u_*^2T}{\Delta t g} - 4.75\overline{w'T'}\right).$$
(B13)

If we return to Eqs. (B9) and (B11), we see that the difference between $h_{e1}$ and $h_e$ is small but unknown. Following the Newton-Rapson method, $h_{e1}$ from Eq. (B11) is used to replace $h_b$, and $h_{e_1}$ on the left side is replaced with a new variable $h_{e_2}$, with its subscript 2 indicating that it is the 2$^{nd}$ approximation for $h_e$. In such a way, $h_e$ can be iteratively approached by $h_{e_{n+1}}$, mathematically described as

$$h_{e_{n+1}} = h_{en} - \frac{f(h_{e_n})}{f'(h_{e_n})}$$
(B14)

where subscript $n$ is a positive integer indicating the $n^{th}$ approximation for $h_e$ while $f(h_{e_n})$ and $f'(h_{e_n})$ can be derived in the same way as for Eqs. (B12) and (B13) from $f(h_x)$. Until $\left|h_{e_{n-1}} - h_{e_{n-1}}\right| < 1.96\sigma$, and as long as $f(h_{e_n})$ and $f'(h_{e_n})$ are valid, $h_e$ can be acquired by

$$h_e \approx h_{e_{n+1}},$$
(B15)

where σ is the published precision of direct measurements from a ceilometer, and 1.96σ is the accuracy in $h_e$ from the
solution procedure above. In this study the precision of the SkyVUE$^{TM}$ PRO Ceilometer is used for σ (5 m, Campbell Scientific Inc., 2025). Alternatively, if during the iteration process, $f(h_{en})$ and/or $f'(h_{en})$ become invalid, $h_e$ can be acquired backwards by

$$h_e \approx h_{e_n}$$
(B16)

And with the value of $h_e$, $h$ value can be calculated from Eq. (B5).

While an eddy-covariance flux system is running into a new $\Delta t$, the $h_e$ value becomes $h_b$ value of current $\Delta t$. However, $h_e$ from a previous $\Delta t$ does not exist in the first $\Delta t$ immediately after an eddy-covariance system starts, or in the case data variables such as $u_*$, $L$, $T$, and/or $\overline{w'T'}$ are not available due to a system restart, power outage, or heavy precipitation/dust interfering with measurements from the sonic anemometer or gas analyzer. In such a case, for quick





starting or resuming the continuity of data, an alternative approach described in the next section can be used to approximate

$h$ for such a "first" output interval.

**B4 Approximation to $h$ for the "first" output interval**

For any "first" $\Delta t$ under neutral to stable conditions, $h$ can be computed from Eq. (B1), and under convective conditions,

it can be approximated from the 2$^{nd}$ order Lagrange interpolation polynomial:

$$h = \left[ \frac{80}{381}(L+30) - \frac{5}{31}(L+15) \right](L+650) + \frac{12}{3937}(L+15)(L+30) , \qquad \text{(B16)}$$

which is developed based on the data from Table 1 in Kljun et al. (2015).  Even after a first $h$ value is obtained, if convective

conditions persist, Eqs. (B7) and (B11) cannot be used until a trend (e.g., at least two values) of $h$ are known.  Once a trend

is established, then the current $h_e$ can be estimated, which can then be substituted for $h_b$ in the next $\Delta t$ over which Eq. (B3)

theory can be applied to estimate $h$ under convective conditions.

**B5 Estimation of $h_b$ every $\Delta t$**

To establish the trend for $h$, at least one more value for $h$ must be acquired in the same way as the "first" $\Delta t$.  Once

known, these values provide an estimate of $h_e$ for current $\Delta t$:

$$h_e = h + \frac{h - h_p}{2} , \qquad \text{(B17)}$$

where subscript $p$ indicates a previous interval.  The estimate becomes $h_b$ value for next $\Delta t$ (i.e., the 3$^{rd}$ $\Delta t$).  This $h_b$ will then

be used to compute $h_e$ from Eqs. (B7) to (B16) under convective boundary layer conditions.  If conditions become neutral to

stable, $h$ is once again directly computed from Eq. (B1) without using $h_b$.

**B6 Summary**

The algorithm developed above was implemented into EasyFlux series (Campbell Scientific Inc. UT, USA) for

computing $h$. The $h$ value is used for the applications of flux footprint equations from Kljun et al. (2015). This value is stored

in flux datasets as the variable name *PBLH_F,* following the Ameriflux variable naming convention (AmeriFlux, 2018).

**Appendix C: Subroutine in EasyFlux for footprint characteristics from Kljun et al. (2015)**

**C1 Variable notation**

| S*ubroutine* | *main program* | |
|---|---|---|
| U_star | USTAR | Friction velocity |
| h_aerodynamic | z | Aerodynamic height |
| Obukhov | MO_LENGTH | Monin-Obukhov length |
| h_PBL | PBLH_F | Planetary boundary layer height |
| u_z | U | Mean wind speed at height of z in the streamwise direction |
| range_intrst | FETCH_INTRST | Upwind fetch of interest (measurement targeted range) |
| FP_range_intrst | FP_FETCH_INTRST | Percentage of measured scalar flux from upwind fetch of interest |
| range(1) | fetch(1) = FETCH_MAX | Upwind location of sources/sinks that contribute most to measured flux |
| range(2) | fetch(2) = FETCH_70 | Upwind fetch within which the sources/sinks contribute 70% to measured flux |
| range(3) | fetch(3) = FETCH_80 | Upwind fetch within which the sources/sinks contribute 80% to measured flux |
| range(4) | fetch(4) = FETCH_90 | Upwind range within which the sources/sinks contribute 90% to measured flux |





**C2 Subroutine**

**Sub Footprnt_Charctrstcs_Kljun_etal2015** (U_star, h_aerodynamic, Obukhov, h_PBL, _
           u_z, range_intrst, FP_range_intrst, rang(4))

**C2.1 Declaration of variables used inside this subroutine**
 In the two-dimensional matrixes below, the 1$^{st}$ row for convective stratifications and the 2$^{nd}$ row for neutral/stable stratifications. The matrixes below symbols are used for code readability.

a. *Equation parameters (a, b, c, and d$_0$) in Table A1 of Kljun et al. (2015)*
          '   a    b    c    d$_0$
Dim paramtr_valus(2, 4) = {2.930, -2.285, 2.127, -0.107, _   'Convective boundary layer stratifications.
        1.472, -1.996, 1.480, 0.169}    'Neutral/stable boundary layer stratifications.
Dim   paramtr_symbls(4)           'Parameter symbols in the model of Kljun et al. (2015).
Alias paramtr_symbls(1) = a
Alias paramtr_symbls(2) = b
Alias paramtr_symbls(3) = c
Alias paramtr_symbls(4) = d0

b. *Index*
Dim stablty_index As Long        'Stratification index: 1 for Obukhov < 0 and 2 for Obukhov >= 0.
Dim i_fp     As Long          'Iteration index for computation.

c. *Matrix for the 1$^{st}$ inflection* $X_{F1}^*$, *maximum* $X_{max}^*$, *and 2$^{nd}$ inflection $X_{F1}^*$ locations on footprint curves (Table 1)*
          ' $X_{F1}^*$      $X_{max}^*$      $X_{F2}^*$
Dim x_star_infl_max_valus(2, 3) = {0.31026689, 0.82385339, 1.3374399, _   'Convective boundary layer stratifications.
760         0.48210189, 0.91048297, 1.3388640}   'Neutral/stable boundary layer stratifications.
Dim x_star_infl_max_symbls(3)         'Symbols for $X^*$ at the inflection and max points.
Alias x_star_infl_max_symbls(1) = x_f1
Alias x_star_infl_max_symbls(2) = x_max
Alias x_star_infl_max_symbls(3) = x_f2

d. *Matrix for cumulative footprint (%) to the end of each characteristic zone (Table 2)*
          ' $X_{F1}^*$      $X_{max}^*$      $X_{F2}^*$
Dim cumul_fp_segmnt_valus(2, 3) = {1.1321783, 15.737952, 32.526482, _   'Convective boundary layer stratifications.
        0.87452260, 13.472100, 28.018350}   'Neutral/stable boundary layer stratifications.
Dim   cumul_fp_segmnt_symbls(3)         'Symbols for the cumulative footprint.
Alias cumul_fp_segmnt_symbls(1) = cumul_x_f1
Alias cumul_fp_segmnt_symbls(2) = cumul_x_max
Alias cumul_fp_segmnt_symbls(3) = cumul_x_f2

e. *Matrix for nondimensional upwind fetches of sources/sinks contributing 70, 80, or 90% to fluxes*
          ' $X_{70}^*$     $X_{80}^*$     $X_{90}^*$      'A subscript indicates percent.
Dim x_star_p_valus(2, 3) = {3.7400033, 5.5734341, 10.371083, _     'Convective boundary layer stratifications.
        4.3702906, 6.9142010, 14.612024}      'Neutral/stable boundary layer stratifications.
Dim   x_star_p_symbls(3)           'Symbols for the fetches.
Alias x_star_p_symbls(1) = x_70
Alias x_star_p_symbls(2) = x_80
Alias x_star_p_symbls(3) = x_90

f. *Working variables*
Variables for computations of *FP_FETCH_INTRST*
Dim x_star_intrst       'Nondimensional upwind fetch of interest for measurements
Dim fp_segmnt_ahead     'Cumulative footprint
Dim x_star         '$X^*$ nondimensional upwind distance to an eddy covariance flux station
Dim integrtn_incrmnt     'Increment for the numerical integration

Variables for use in Composite Simpson's Rule for numerical integrations
Dim FP_start        'Footprint value at the starting $X^*$ of integration section
Dim FP_odd        'Summed values of footprint at $X^*$ on the right boundary of sequentially odd increment





Dim FP_even          'Summed values of footprint at $X^*$ on the right boundary of sequentially even increment
Dim FP_end          'Footprint at the ending $X^*$ of integration section

**C2.2 Computations**

*a. Variable Preparation*

```
Select Case Obukhov
   Case Is < 0
      stablty_index = 1
      Move (paramtr_symbls(1),           4, paramtr_valus(1, 1),          4)
      Move (x_star_infl_max_symbls(1),  3, x_star_infl_max_valus(1, 1),  3)
      Move (cumul_fp_segmnt_symbls(1), 3, cumul_fp_segmnt_valus(1, 1), 3)
      Move (x_star_p_symbls(1),          3, x_star_p_valus(1, 1),         3)

   Case Is >= 0
      stablty_index = 2
      Move (paramtr_symbls(1),           4, paramtr_valus(2, 1),          4)
      Move (x_star_infl_max_symbls(1),  3, x_star_infl_max_valus(2, 1),  3)
      Move (cumul_fp_segmnt_symbls(1), 3, cumul_fp_segmnt_valus(1, 1), 3)
      Move (x_star_p_symbls(1),          3, x_star_p_valus(2, 1),         3)

EndSelect 'Obukhov
```

*b. FETCH_MAX*

rang(1) = (k*x_max*h_aerodynamic*u_z)/(U_star*(1-h_aerodynamic/h_PBL))   'k is van Karman constant, given in main program

*c. FETCH_70, FETCH_80, and FETCH_90*

```
rang(2) = (k*x_70*h_aerodynamic*u_z)/(U_star*(1-h_aerodynamic/h_PBL))
rang(3) = (k*x_80*h_aerodynamic*u_z)/(U_star*(1-h_aerodynamic/h_PBL))
rang(4) = (k*x_90*h_aerodynamic*u_z)/(U_star*(1-h_aerodynamic/h_PBL))
```

*d. Footprint portion of measured flux within an upwind fetch of interest for measurements in real-scale fields*

Preparation for numerical integration

x_star_intrst = (range_intrst/h_aerodynamic)*(1-h_aerodynamic/h_PBL)*(U_star/(k*u_z))

```
Select Case x_star_intrst
   Case Is <= x_f1
      integrtn_incrmnt  = (x_star_intrst - d0*(1+1e-7))/1000
      fp_segmnt_ahead  = 0
      x_star              = d0*(1+1e-7)

   Case Is > x_f1 AND Is <= x_max
      integrtn_incrmnt  = (x_star_intrst - x_f1)/1000
      fp_segmnt_ahead = cumul_x_f1
      x_star = x_f1

   Case Is > x_max AND Is <= x_f2
      integrtn_incrmnt  = (x_star_intrst - x_max)/1000
      fp_segmnt_ahead = cumul_x_max
      x_star = x_max

   Case Is > x_f2
      integrtn_incrmnt  = (x_star_intrst - x_f2)/1000
      fp_segmnt_ahead = cumul_x_f2
      x_star = x_f2

EndSelect 'x_star_intrst
```

Preliminary values of FP_start, FP_odd, FP_even for use inside an iteration

FP_start = (a*(x_star - d0)^b)*EXP(-c/(x_star - d0))                 'Footprint at the starting $X^*$ of integration section
FP_odd  = 0
FP_even = 0



```
For i_fp = 1 To 499

    x_star   = x_star + integrtn_incrmnt
    FP_odd = FP_odd + (a*(x_star - d0)^b)*EXP(-c/(x_star - d0))

    x_star   = x_star + integrtn_incrmnt
    FP_even = FP_even + (a*(x_star - d0)^b)*EXP(-c/(x_star - d0))

Next i_fp
FP_end   = (a*(x_star - d0)^b)*EXP(-c/(x_star - d0))
FP_even = FP_even - FP_end
```

Composite Simpson's Rule for numerical integrations (below, the 2^nd term on the right)
FP_range_intrst = fp_segmnt_ahead + 100*(integrtn_incrmnt/3)*(FP_start + 4*FP_odd + 2*FP_even + FP_end)

EndSub 'Footprnt_Charctrstcs_Kljun_etal2015

**C3 The use of subroutine in the main program of EasyFlux series (Campbell Scientific Inc., UT, USA)**
The Subroutine to compute the footprint characteristics from Kljun et al. (2015) is used in EasyFlux series through a Call instruction:
Call Footprnt_Charctrstcs_Kljun_etal2015(*USTAR*, *z*, *MO_LENGTH*, *PBLH_F*,
$\qquad\qquad\qquad$ *U*, *FETCH_INTRST*, *FP_FETCH_INTRST*, *fetch*(1))

For every averaging interval in eddy-covariance systems, *USTAR*, *z*, *MO_LENGTH*, and *U* have their values available from measurements, *PBLH_F* is computed using the algorithm from Appendix B, *FETCH_INTRST* is entered by a user before or while EasyFlux is running, and the values of flux footprint characteristics are output from this subroutine above that is executable as long as a prime is put ahead of each line of text.

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
