# Peer review of "Application of flux footprint equations from Kljun et al. (2015) to field eddy-covariance systems for footprint characteristics into flux network datasets"

_EGUsphere, 2025_

## Community Comment (CC12)

**Response to**
https://doi.org/10.5194/egusphere-2025-4576-CC11

(*See citations for responses in the posted manuscript*)

This is a very detailed study on the application of the footprint model. It would be very useful to users of the Campbell Eddy Covariance Systems. One particular point that is important in my opinion is that the limitations of the Kljun et al. (2015) model should be explicitly discussed. Further clarifications at some places may also be important. The specific comments are provided below.

**Response**:
Thank you so much for your professionally detailed comments.  We appreciate your high expectations of some comments beyond the scope of this work. Overall, your comments remind us of clarifying some points in further revision.

The major objective of this study is to optimize field computations balancing time and accuracy for footprint characteristics from the analytical footprint equations commonly adopted by the community over last 20 years while using visualization like Fig. 1 to help non-expert readers easily understand footprint and also extending the applications of non-dimensional footprint equations from Kljun et al. (2015) for eddy-covariance system setting (e.g., measurement height determination). Kljun et al (2004) had some limitations as mentioned on page 512. Their application is limited to the following ranges of atmospheric stability, friction velocity, and measurement height:

a. $-200 \leq (z_m - d)/L \leq 1$
b. $u_* \geq 0.2$
c. $z_m - d \geq 1\,\mathrm{m}$

(2)

where $z_m$ is measurement height, $d$ is zero displacement height, $L$ is Obukhov length, and $u_*$ is friction velocity. These limitations are not an issue any more in Kljun et al. (2015) (personal communication, Dr. Kljun).

Line 17 and some lines later. A footprint is a transfer function relating the source area to the measured flux. The footprint itself is not an area. However, the footprint can be used to calculate the footprint. See Steinfeld et al. (2008) and Fu et al. (2025). In addition, even over a horizontally homogeneous terrain, it is likely that the footprint can extend to the downwind side of the EC system. Over complex terrain, as found by Fu et al. (2025), the extension in the downwind direction may be substantial.

References:
Steinfeld, G., Raasch, S., & Markkanen, T. (2008). Footprints in homogeneously and heterogeneously driven boundary layers derived from a Lagrangian Stochastic Particle Model embedded into large-eddy simulation. Boundary-Layer Meteorology, 129(2), 225–248. https://doi.org/10.1007/s10546-008-9317-7

Fu, S., Chen, J. M., Zhang, J., Cheng, Z., Miao, G., Wang, R., et al. (2025). Flux footprints over a forested hill derived from a Lagrangian particle model coupled into a large-eddy simulation model. Journal of Geophysical Research: Atmospheres, 130, e2025JD043591. https://doi.org/10.1029/2025JD043591

**Response**:
"Footprint" here can be interpreted as probability density or the area over which a sensor sensed, depending on the context. The unit of conventional crosswind-integrated footprint is $m^{-1}$. This unit derived from $m^{-2}$ over 2D field, implicitly indicating the footprint is related to an area.

The flux footprint can be extended to downwind area in extreme cases under boundary-layer advection conditions (i.e., very low wind speed). The figures in this manuscript just used equations of Kormann and Meixner (2001) and Kljun et al. (2015). The advancement in Fu et al. (2025) beyond the analytical equations was published in Sept 2025 before the submission of this manuscript. We will learn and cite this article when we further revise our manuscript.

Lines 25-29. It is unclear why the model can be efficiently calculated in the field. From the summary section, it becomes clear that some parts are precalculated so it is not necessary to calculate them in the field. I suggest adding this explicitly into the abstract.

**Response**:
Footprint characteristics are required to be included in the submission datasets to flux networks, such ChinaFlux, AmeriFlux (2018), and FluxNet. It would be huge time saving if the datasets directly from field could be submitted to a network. We will further clarify this point in further revision.

Lines 30-31. Since the footprint model by Kljun et al. (2015) was developed for flat ground, it may not be appropriate to state that the model developed based on Kljun et al. (2015) can be used over complex terrain.

**Response**:
"The applicability of Kljun et al. (2015) to for complex terrains was not mentioned although we said "*This computational approach may also be applied to footprint analyses over complex terrain, nonuniform sources/sinks, or in cases where other footprint equations are used*."

Lines 63-75. The discussion of the symmetry seems to suggest that the source area for a negative flux is in the downwind side of the EC system. This is in contrast to my expectation.

**Response**
Negative flux through the measurement volume would be transferred to downwind sinks, instead of sources. This flux footprint should be symmetric with its upwind counterpart.

Section 4.4. Since the boundary layer height appears at multiple places in the footprint model. It might be important to the users how sensitive the footprint model is to the boundary layer height. In addition, it might be informative to the users if some examples of the boundary layer height are calculated and provided using methods in Appendix B, and these example heights can be compared to the typical values from the literatures.

**Response**
Sensitivity tests are the scope of development of original footprint equations (reference to section 3 in Kljun et al. 2015). This reference should be mentioned in our revision.